# Equivalence and its invalidation between non-Markovian and Markovian spreading dynamics on complex networks

Mi Feng [1,2,3], Shi-Min Cai[2,3], Ming Tang [1,4] & Ying-Cheng Lai [5]

Epidemic spreading processes in the real world depend on human behaviors and, consequently, are typically non-Markovian in that the key events underlying the spreading dynamics cannot be described as a Poisson random process and the corresponding event time is not exponentially distributed. In contrast to Markovian type of spreading dynamics for which mathematical theories have been well developed, we lack a comprehensive framework to analyze and fully understand non-Markovian spreading processes. Here we develop a mean-field theory to address this challenge, and demonstrate that the theory enables accurate prediction of both the transient phase and the steady states of non-Markovian susceptible-infected-susceptible spreading dynamics on synthetic and empirical networks. We further find that the existence of equivalence between non-Markovian and Markovian spreading depends on a specific edge activation mechanism. In particular, when temporal correlations are absent on active edges, the equivalence can be expected; otherwise, an exact equivalence no longer holds.

[1] School of Mathematical Sciences, Shanghai Key Laboratory of PMMP, East China Normal University, Shanghai 200241, China. [2] Web Sciences Center, University of Electronic Science and Technology of China, Chengdu 611731, China. [3] Big Data Research Center, University of Electronic Science and Technology of China, Chengdu 611731, China. [4] Shanghai Key Laboratory of Multidimensional Information Processing, East China Normal University, Shanghai 200241, China. [5] School of Electrical, Computer and Energy Engineering, Arizona State University, Tempe, AZ 85287, USA. Correspondence and requests for materials should be addressed to M.T. (email: tangminghan007@gmail.com)

D isease or virus spreading on complex networks, because of its broad relevance to health care, social, economical, and political sciences as well as information technologies, has been an active area of research on contemporary network science and engineering[1–8]. Traditional models of spreading dynamics on networks are memoryless Markovian in the sense that, for all individuals in the network, both contracting a virus and recovering from it are viewed as a Poisson process. The time interval between two successive contracting events and that between two adjacent recovery events follow an exponential distribution with a constant rate—the arrival rate of the respective Poisson process. Associated with the exponential distribution is the memoryless property: any future waiting time does not depend on the previous waiting time. This property fits with that of a memoryless Markovian process as, what matters in order to predict the future is the current state, not the history of the process. The Markovian assumption greatly facilitates the development of mathematical theories of spreading process on complex networks[5,7] through, e.g., a mean-field type of analyses[1,9–11] of the standard susceptible-infected-susceptible (SIS) or susceptible-infected-recovered (SIR) process. There has been increasing empirical evidence and modeling effort that the occurrences of contacts associated with human activities are non-Markovian temporal processes with a heavy tailed inter-event time distribution[12–33]. A Markovian description of network spreading dynamics is thus ideal and provides only an approximate picture of the real world.

The past decade has witnessed a growing interest in non-Markovian spreading dynamics on complex networks[34–50]. The failure of the Markovian framework in describing human interactions in relation to disease spreading was noted quite early, and it was found that the deviation from the exponential distribution of the inter-event time to being heterogeneous can impede spreading[35]. A non-Markovian SIR model with arbitrary time distributions of infection and recovery was solved through the approach of dynamical message passing[37]. It was also found that a heavy-tailed waiting time distribution can slow down the prevalence decay[39]. An SIR model with fixed recovery time but with a heavy-tailed infection time distribution was studied with the finding that temporal heterogeneity in the contact process can significantly suppress epidemic spreading[40]. A relatively significant alteration of the outbreak threshold was reported for non-Markovian type of SIS infection events and certain equivalence between non-Markovian and Markovian models through redefining the effective infection rate was pointed out[41,42]. Two basic dynamical rules governing a non-Markovian type of SIS spreading process were uncovered[43]. More recently, a method was proposed to estimate the effective infection rate for non-Markovian type of spreading dynamics[47].

A common theoretical tool to deal with network spreading dynamics is mean-field analysis. The earlier version of the mean-field theory assumed that all nodes in the network are regarded as statistically equivalent[9]. To account for the non-homogeneous nature of real world networks, a heterogeneous mean field theory was developed in which the nodes with the same degree are considered as equivalent[1]. A more systematic approach to fully capturing the network topology was articulated—the so-called quench mean-field approximation[10,11]. To take into account dynamic correlations, the method of pairwise approximation can be exploited[51–54] which centers about analyzing the evolution of the states of nodal pairs. An approximate master equation theory can lead to more accurate theoretical predictions and can be reduced to the pair-wise approximation theory and the mean field theory through proper approximations[55]. A mean-field analysis based on the pair approximation for non-Markovian SIR dynamics on networks was recently developed[45,46,48].

Our work is motivated by two considerations. The first concerns about the development of a general theoretical framework to deal with non-Markovian processes. In particular, in spite of the previous studies, a comprehensive framework to analyze and understand the full dynamical evolution of non-Markovian spreading processes is still lacking. Especially, in the existing literature on network spreading dynamics, Markovian or non-Markovian, a central focus has been on the final steady state of the system, with the transient dynamics leading to the final state largely ignored. In nonlinear dynamical systems, transient behaviors have been recognized as relevant as or even more important than the final state. For example, transient chaos arises commonly in nonlinear dynamical systems and is typically more ubiquitous than attractors (final states)[56]. In ecological systems, transient dynamics have been long recognized as the main source on which empirical observations rely and are thus a key driving force of natural evolution[57–61]. For non-Markovian type of network spreading dynamics, we lack a theory to describe both transient dynamics and final steady state. The second motivation is that, since Markovian spreading processes, while ideal, are amenable to rigorous mathematical analyses and thus often afford a complete understanding of the detailed underlying dynamics, it is of great interest to uncover conditions under which a non-Markovian process is equivalent to or can be approximated by a Markovian one.

In this paper, we articulate a first-order mean field theory for SIS dynamics with non-Markovian infection and recovery processes. We define the probability density function of state age of node (or of edge) and derive the corresponding partial differential equations that govern the evolution of the density functions. We demonstrate that the theory can predict well both the transient behaviors and the steady states of non-Markovian spreading on synthetic and empirical networks. A key finding is that the edge activation rule determines whether there is an equivalence between non-Markovian and Markovian SIS dynamics, and we show that a specific type of activation rule can lead to an exact equivalence. For non-Markovian processes that are not equivalent to Markovian ones, a relatively large infection density of the whole network makes the process closer to being Markovian. With the ability to analyze transient dynamics, the developed theoretical framework not only leads to a picture of the entire evolution process of non-Markovian spreading dynamics, but also results in a deeper understanding of the conditions under which a non-Markovian process is equivalent to a Markovian one.

## Results

**Non-Markovian spreading dynamics on complex networks**. In the classical, Markovian SIS model, a node in a network can be in the susceptible or infected state. An infected node can pass on the virus to any of its susceptible neighbors at certain transmission rate and return to the pool of susceptible nodes at a fixed recovery rate. Because of the Markovian and memoryless nature of the model, transmission of disease and recovery of nodes both are Poisson processes, i.e., both the infection and recovery time are exponentially distributed. In contrast, in a non-Markovian epidemic process with memory, neither the infection nor the recovery time follows an exponential distribution. Instead, such distributions are typically "fat-tailed" due to the highly heterogeneous nature of various human behaviors[12–33].

For a non-Markovian epidemic process, the infection and recovery rates are generally time dependent. To gain insights into defining these rates, we begin with a key quantity: the infection age (or susceptibility age) $\tau$ of a node, which is the time elapsed from the birth of its current state, i.e., an infected state or a

susceptible state, to the current time $t$. If a node becomes infected at time $t - \tau$ and has not recovered by time $t$, its infection age at time $t$ will be $\tau$. An infected node decays spontaneously into the susceptible state after a random time $\tau$ whose probability distribution is $\psi_{rec}(\tau)$, indicating that the infected node will recover during the infinitesimal infection age interval $(\tau, \tau + d\tau)$ with the probability $\psi_{rec}(\tau)d\tau$. Similarly, the activation age (or non-activation age) $\kappa$ of an edge can be defined, where the active edges host statistically independent stochastic infection processes of the same distribution $\psi_{inf}(\kappa)$. That is, the probability that an active edge transmits the disease during the infinitesimal active age interval $(\kappa, \kappa + d\kappa)$ is $\psi_{inf}(\kappa)d\kappa$. If a susceptible node has more than one active edge, an infection process will take place independently along each active edge.

The time dependent recovery and infection rates can then be evaluated[43,47] based on the distributions $\psi_{rec}(\tau)$ and $\psi_{inf}(\kappa)$. In particular, if an event has not taken place by a time since the process was initiated, it will take place in the next time interval with a conditional probability. The recovery and infection rates are thus given by, respectively,

$$\omega_{rec}(\tau) = \frac{\psi_{rec}(\tau)}{\Psi_{rec}(\tau)} \qquad (1)$$

and

$$\omega_{inf}(\kappa) = \frac{\psi_{inf}(\kappa)}{\Psi_{inf}(\kappa)}, \qquad (2)$$

where $\Psi_{rec}(\tau)$ and $\Psi_{inf}(\kappa)$ are the corresponding survival probabilities. Especially,

$$\Psi_{rec}(\tau) = \int_{\tau}^{+\infty} \psi_{rec}(\tau')d\tau' \qquad (3)$$

is the probability that the infected node will not recover before the infection age $\tau$ and

$$\Psi_{inf}(\kappa) = \int_{\kappa}^{+\infty} \psi_{inf}(\kappa')d\kappa' \qquad (4)$$

is the probability that the active edge never transmits the disease in the range of the active age from 0 to $\kappa$. Substituting $\Psi_{rec}(\tau)$ and $\Psi_{inf}(\kappa)$ into Eqs. (1) and (2), we obtain

$$\Psi_{rec}(\tau) = e^{-\int_0^{\tau} \omega_{rec}(\tau')d\tau'} \qquad (5)$$

and

$$\Psi_{inf}(\kappa) = e^{-\int_0^{\kappa} \omega_{inf}(\kappa')d\kappa'} \qquad (6)$$

The recovery and infection time distributions can thus be expressed, respectively, as

$$\psi_{rec}(\tau) = \omega_{rec}(\tau)e^{-\int_0^{\tau} \omega_{rec}(\tau')d\tau'}, \qquad (7)$$

and

$$\psi_{inf}(\kappa) = \omega_{inf}(\kappa)e^{-\int_0^{\kappa} \omega_{inf}(\kappa')d\kappa'}. \qquad (8)$$

In the special case of memoryless Markovian model, the temporal processes follow the Poisson statistics, where the distributions are exponential with their respective constant rate.

For epidemic spreading on a network, there are various mechanisms to activate edges. We focus on the two basic mechanisms to generate active edges[43,47]. In general, an undirected edge can be regarded as being equivalent to two directed edges in the opposite directions, and a directed edge with starting node $j$ and ending node $i$ is denoted as $i \leftarrow j$. For the first mechanism (type-I), which is the same as rule 1 in ref. [43], one defines a directed link $i \leftarrow j$ as an active edge when node $j$ is an infected node and node $i$ is susceptible. If an active edge $i \leftarrow j$

transmits the disease, the healthy node $i$ will turn into the infected state and the edge $i \leftarrow j$ will become non-active. For the second mechanism (type-II), which is the same as that in ref. [47], a directed link $i \leftarrow j$ is defined as an active edge when node $j$ is an infected node regardless of the state of node $i$. Once the active edge $i \leftarrow j$ transmits the disease, node $i$ will turn into the infected state if it is susceptible, or nothing happens to node $i$ if it is already infected. At the same time, the active edge $i \leftarrow j$ will become a new active edge with active age $\kappa = 0$. Figure 1 shows the cases in which the active age of an active edge is zero. In the special case of non-Markovian process where $\psi_{inf}(\kappa)$ follows an exponential distribution and the infection rate $\omega_{inf}(\kappa)$ is a constant independent of time, the SIS models constructed according to the two respective mechanisms are equivalent.

**First-order mean-field theory for non-Markovian spreading.** To describe the full dynamical evolution of non-Markovian epidemic spreading on networks, we articulate a theoretical framework based on the approach of first-order mean-field analysis. Specifically, we begin by defining $I_i(\tau; t)$ and $S_i(\tau; t)$ as the probability density functions that node $i$ stays in the infected and susceptible state aged $\tau$ at time $t$, respectively[48]. The probability of node $i$ being in the infected state aged from $\tau$ to $\tau + d\tau$ is thus $I_i(\tau; t)d\tau$. In the time interval $(t, t + dt)$, node $i$ returns to the susceptible state with probability $\omega_{rec}(\tau)dt$ and the probability that the state of the node remains unchanged is $1 - \omega_{rec}(\tau)dt$. After the infinitesimal time interval $dt$ has elapsed, the age of node $i$ and that of the active edge $i \leftarrow j$ both will increase by the amount $dt$. We thus have $dt = d\tau = d\kappa$. At time $t + dt$, the probability density function that node $i$ still remains in the infected state aged $\tau + d\tau$ is given by

$$I_i(\tau + d\tau; t + dt) = [1 - \omega_{rec}(\tau)d\tau]I_i(\tau; t). \qquad (9)$$

This difference equation can be rewritten as a partial differential equation:

$$\left(\frac{\partial}{\partial \tau} + \frac{\partial}{\partial t}\right)I_i(\tau; t) = -\omega_{rec}(\tau)I_i(\tau; t). \qquad (10)$$

Since the infected node $i$ with infection age ranging from 0 to $+\infty$ can switch into the susceptible state of age zero insofar as there is a recovery, the probability density function that node $i$ returns to the susceptible state aged $\tau = 0$ is

$$S_i(0; t + dt) = \int_0^{+\infty} \omega_{rec}(\tau)I_i(\tau; t)d\tau. \qquad (11)$$

To describe the time evolution of $S_i(\tau; t)$, we assume that the ages of two connected nodes are uncorrelated. The probability density function that node $i$ in the susceptible state of age $\tau$ is infected by node $j$ at time $t$ can be written as $\Phi_{i \leftarrow j}(\tau; t)$. During the time interval $(t, t + dt)$, the susceptible node $i$ aged $\tau$ is infected by its neighbors with the probability $\sum_{j=1}^{N} a_{ij}\Phi_{i \leftarrow j}(\tau; t)d\tau$, where $N$ is the network size and $a_{ij}$ is the $ij$th element of the network adjacency matrix. We thus get the partial differential equation governing the evolution of $S_i(\tau; t)$ as

$$\left(\frac{\partial}{\partial \tau} + \frac{\partial}{\partial t}\right)S_i(\tau; t) = -S_i(\tau; t)\sum_{j=1}^{N} a_{ij}\Phi_{i \leftarrow j}(\tau; t). \qquad (12)$$

Since a susceptible node $i$ with any susceptibility age ranging from 0 to $+\infty$ can be infected and switches into the infected state aged zero as a result of the infection process, the probability density

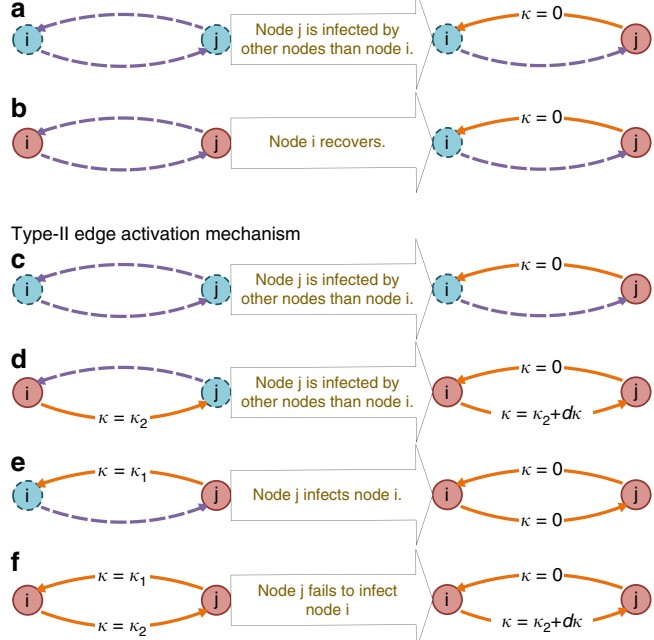

**Type-I edge activation mechanism**

**a** Node j is infected by other nodes than node i.

**b** Node i recovers.

**Type-II edge activation mechanism**

**c** Node j is infected by other nodes than node i.

**d** Node j is infected by other nodes than node i.

**e** Node j infects node i.

**f** Node j fails to infect node i

**Fig. 1** Two mechanisms to activate edges. For the first mechanism (type-I), there are two cases in which the state age of an active edge is set to zero: **a** the event is driven by the generation of infected node, **b** it results from the generation of an susceptible node. For the second mechanism (type-II), there are four independent scenarios for setting zero the age of an active edge: **c**, **d** the event occurs because of the generation of an infected node, **e**, **f** it is caused by the transmission of the disease. Blue dashed and red solid circles represent susceptible and infected nodes, respectively. Purple dashed and orange solid arrows denote nonactive and active edges, respectively. Each state transition takes place in an infinitesimal time interval $d\tau$ or $d\kappa$, where $d\tau = d\kappa$

function that node $i$ enters into the infected state aged $\tau = 0$ is

$$I_i(0; t + dt) = \int_0^{+\infty} S_i(\tau; t) \sum_{j=1}^N a_{ij} \Phi_{i \leftarrow j}(\tau; t) d\tau. \qquad (13)$$

For different activation mechanisms of active edges, namely, type-I and type-II, the forms of $\Phi_{i \leftarrow j}(\tau; t)$ are different. For type-I mechanism, once an infected node $j$ and a susceptible node $i$ appear on both ends of an edge, the directed edge $i \leftarrow j$ will be activated or, equivalently, the edge will enter into an active state aged $\kappa = 0$. Ignoring the dynamical correlation between any pairwise nodes, the age of the active edge is determined by the smaller age of the connected nodes, $\min(\tau, \tau')$, where $\tau$ and $\tau'$ are the susceptibility age of node $i$ and the infection age of node $j$, respectively. We thus have

$$\Phi_{i \leftarrow j}(\tau; t) = \int_0^{+\infty} \omega_{\text{inf}}[\min(\tau, \tau')] I_j(\tau'; t) d\tau', \qquad (14)$$

where $\omega_{\text{inf}}[\min(\tau, \tau')]$ is the infection rate of the active edge whose age is equal to the smaller one of the end nodes.

For type-II mechanism, the age of the active edge $i \leftarrow j$ depends on the infected node $j$ only, and we have

$$\Phi_{i \leftarrow j}(\tau; t) = \int_0^{+\infty} \eta(\tau') I_j(\tau'; t) d\tau', \qquad (15)$$

where the infection rate $\eta(\tau)$ of the infected node $j$ aged $\tau$ satisfies

the integral equation

$$\eta(\tau) = \int_0^\tau \eta(\tau') \psi_{\text{inf}}(\tau - \tau') d\tau' + \psi_{\text{inf}}(\tau). \qquad (16)$$

The solution of Eq. (16) can be written as

$$\eta(\tau) = \int_{\sigma-i\infty}^{\sigma+i\infty} \frac{\hat{\psi}_{\text{inf}}(s)}{1 - \hat{\psi}_{\text{inf}}(s)} e^{s\tau} ds, \qquad (17)$$

where $\hat{\psi}_{\text{inf}}(s)$ is the Laplace transform of $\psi_{\text{inf}}(\tau)$:

$$\hat{\psi}_{\text{inf}}(s) = \int_0^{+\infty} \psi_{\text{inf}}(\tau) e^{-s\tau} d\tau \qquad (18)$$

Supplementary Fig. 1 in Supplementary Note 2 presents the analysis of $\eta(\tau)$ for type-II activation mechanism.

Equations (10) and (12) govern the time evolution of non-Markovian SIS spreading dynamics in general, with the boundary conditions given by Eqs. (11) and (13). The state transition processes can be described as the probability flows between the infected and susceptible states determined by the functions $I_i(\tau; t)$ and $S_i(\tau; t)$, as well as those within each state. A schematic illustration of the various probability flows is shown in Fig. 2. The initial conditions for Eqs. (10) and (12) are the initial probability distributions of each node:

$$I_i(\tau; 0) = \rho_i(\tau) \qquad (19)$$

and

$$S_i(\tau; 0) = \chi_i(\tau), \qquad (20)$$

where $\rho_i(\tau)$ and $\chi_i(\tau)$ are the probability densities of the ages of node $i$ being in the infected and susceptible state, respectively.

**Transient behaviors**. To test the power of our mean-field theory to predict transient behaviors in non-Markovian processes, we carry out Monte Carlo simulations[47,62] of SIS dynamics on different types of networks (see "Methods" for details). To be general, in the simulations, we set $\psi_{\text{inf}}(\kappa)$ and $\psi_{\text{rec}}(\tau)$ to be the long-tailed, Weibull type of distribution as

$$\psi_{\text{inf}}(\kappa) = \frac{\alpha_{\text{I}}}{\beta_{\text{I}}} \left(\frac{\kappa}{\beta_{\text{I}}}\right)^{\alpha_{\text{I}}-1} e^{-\left(\frac{\kappa}{\beta_{\text{I}}}\right)^{\alpha_{\text{I}}}} \qquad (21)$$

and

$$\psi_{\text{rec}}(\tau) = \frac{\alpha_{\text{R}}}{\beta_{\text{R}}} \left(\frac{\tau}{\beta_{\text{R}}}\right)^{\alpha_{\text{R}}-1} e^{-\left(\frac{\tau}{\beta_{\text{R}}}\right)^{\alpha_{\text{R}}}}, \qquad (22)$$

where $(\alpha_{\text{I}}, \alpha_{\text{R}})$ are the shape parameters and $(\beta_{\text{I}}, \beta_{\text{R}})$ are scale parameters. A smaller value of the shape parameter and/or a larger value of the scale parameter corresponds to more extensive heterogeneity of the age distribution.

Figure 3 shows, for three different types of networks, the evolution of the infected density of the entire network over time, where the density value predicted by the mean-field theory is calculated with $I(t) = \sum_{i=1}^N I_i(t)/N$. Initially, 1% of the nodes are randomly chosen to be the infection seed and their infection ages are all zero. A small $\alpha_{\text{I}}$ value for which the distribution $\psi_{\text{inf}}(\kappa)$ is strongly heterogeneous will lead to a large-scale disease outbreak, due mainly to the small mean infection time. In Fig. 3a–c, the Monte-Carlo simulation results and the theoretical predictions for the type-I activation mechanism are presented for Erdös-Rényi (ER) random, Barabási-Albert (BA) scale-free, and Hamsterster networks, respectively. The Hamsterster network is a real-world social network in the human society[63]. Figure 3d–f shows the comparison results of the temporal evolution of the infected density for processes with type-II activation mechanism. The mean-field predictions of the time evolution are generally in good agreement with those of non-Markovian type of SIS

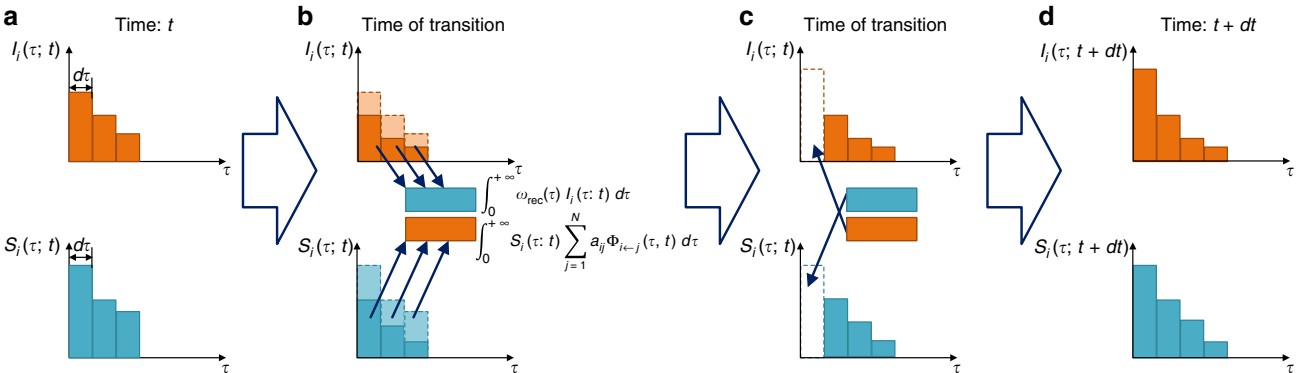

**Fig. 2** Schematic representation of probability flows. Each illustration is divided into blocks of width $d\tau \to 0$, and the area of each block represents the probability that node $i$ is in the corresponding state aged from $\tau$ to $\tau + d\tau$. **a** Two distributions, $I_i(\tau; t)$ and $S_i(\tau; t)$, of node i at time $t$. **b** Each block losses probability $\omega_{\text{rec}}(\tau)I_i(\tau; t)d\tau$ or $S_i(\tau; t)\sum_{j=1}^{N} a_{ij}\Phi_{i\leftarrow j}(\tau; t)d\tau$ and converges to $\int_0^{+\infty} \omega_{\text{rec}}(\tau)I_i(\tau; t)d\tau$ and $\int_0^{+\infty} S_i(\tau; t)\sum_{j=1}^{N} a_{ij}\Phi_{i\leftarrow j}(\tau; t)d\tau$, respectively. **c** Each block moves one step to the right, and the empty blocks $S_i(0; t+dt)$ and $I_i(0; t+dt)$ are filled with the probabilities $\int_0^{+\infty} \omega_{\text{rec}}(\tau)I_i(\tau; t)d\tau$ and $\int_0^{+\infty} S_i(\tau; t)\sum_{j=1}^{N} a_{ij}\Phi_{i\leftarrow j}(\tau; t)d\tau$, respectively. **d** The distributions $I_i(\tau; t+dt)$ and $S_i(\tau; t+dt)$ at time $t+dt$

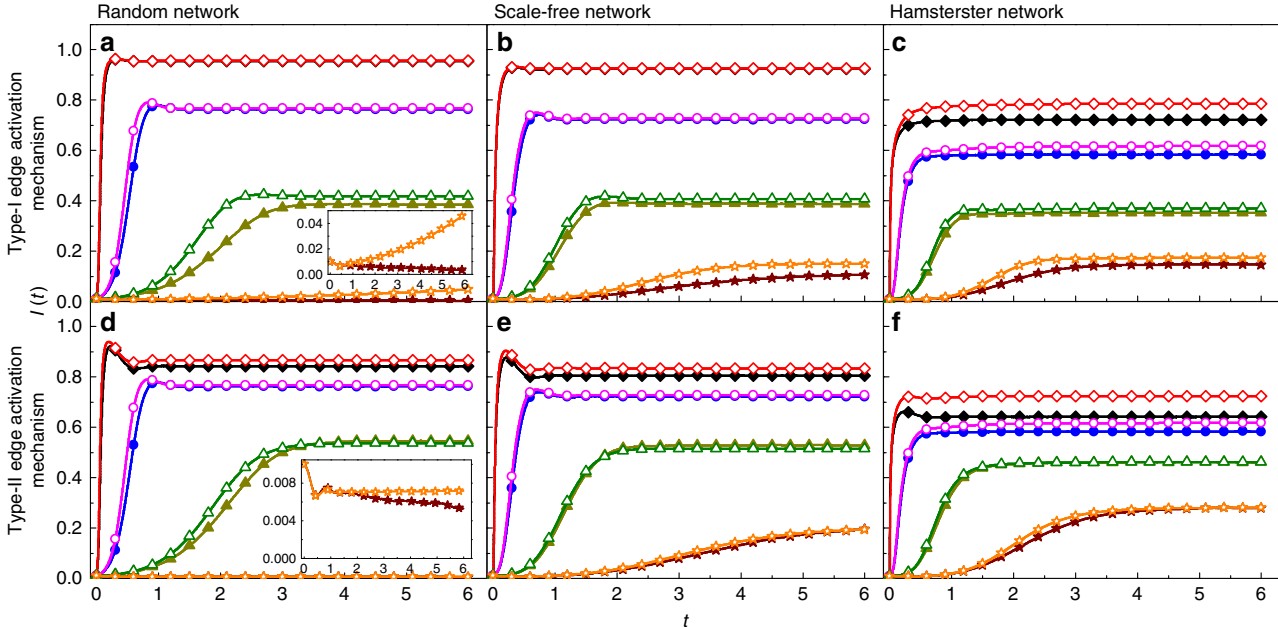

**Fig. 3** (Color online) Comparison of time evolution between simulated and theoretical results. Panels in the first and second rows (**a**–**c** and **d**–**f**), respectively, are for the type-I and type-II mechanisms. The three columns from left to right are for ER random, BA scale-free, and Hamsterster networks, respectively. In all panels, the solid symbols represent the results of simulations which are averaged over 100 realizations for random and scale-free networks, and over 400 realizations for the Hamsterster network. The open symbols represent the results of theoretical solutions obtained from Eqs. (10)–(13). The symbols diamonds, circles, triangles, and stars correspond to $\alpha_I = 0.5, 1, 2, 4$, respectively. The random and scale-free networks have size $N = 10^4$ and mean degree $\langle k \rangle \approx 10$, and the Hamsterster network has $N = 2426$ and $\langle k \rangle \approx 13.7$. The insets in panels **a**, **d** show the extinction process with $\alpha_I = 4$. Other parameters are $\beta_I = 1$, $\alpha_R = 2$, and $\beta_R = 0.5$

spreading dynamics on random (homogeneous), scale-free (heterogeneous), and Hamsterster (real-world) networks.

However, in some cases (e.g., $\alpha_I = 4$ for ER random networks), there are some discrepancies between the predictions from the first-order mean-field theory and the simulation results [c.f., insets in Fig. 3a, d], due to the exclusion of any dynamical correlation. For a more accurate description of non-Markovian spreading processes, the dynamical correlation in the evolution of states of connected nodal pairs must be taken into account. To

meet this challenge, we articulate a second-order mean field approach and show that it is capable of predicting the simulation results more accurately than the first-order theory, even for extreme situations where the disease decays rapidly [e.g., the $\alpha_I = 4$ case in Supplementary Fig. 2c, d]. The second-order theory indicates that, in general, dynamical correlation such as temporal correlation between active edges can significantly affect the accuracy of the mean-field analysis (see Supplementary Fig. 2 and Supplementary Note 3 for details). Due to the limitation of

computational feasibility, we have tested the predictions of the second-order mean field theory for homogeneous networks. To extend the study to heterogeneous networks is currently infeasible due to the extreme high computational complexity required to treat all possible nodal pairs separately.

We further investigate the effects of infection time distribution and degree distribution on the transient time in Supplementary Notes 4 and 5, respectively. For a certain network topology and a given edge activation mechanism, a larger value of the shape parameter $\alpha_I$ leads to a longer transient phase. Intuitively, a narrower distribution of the infecting activities makes it easier for the system to reach a final steady state. (Supplementary Fig. 3 displays results on transient time versus $\alpha_I$). For fixed values of the structural and dynamical parameters, random networks lead to longer transients, due mainly to the lack of hub nodes that can act as "super-spreaders". We find that a smaller value of the power-law exponent and a larger value of the lower-cutoff degree can lead to a shorter transient lifetime, indicating that hub nodes in a scale-free network can expedite spreading processes of the non-Markovian type. Supplementary Fig. 4 presents systematic results on the transient lifetime versus the values of the power-law exponent and the lower-cutoff degree.

**Equivalence between non-Markovian and Markovian spreading.** To establish the conditions under which an equivalence between non-Markovian and Markovian type of SIS spreading dynamics arises, we analyze the steady-state behavior. For non-Markovian type of SIS spreading dynamics for $t \to +\infty$, we define the following asymptotic probability density functions:

$$\tilde{I}_i(\tau) = \lim_{t \to +\infty} I_i(\tau; t), \tag{23}$$

$$\tilde{S}_i(\tau) = \lim_{t \to +\infty} S_i(\tau; t), \tag{24}$$

$$\tilde{\Phi}_{i \leftarrow j}(\tau) = \lim_{t \to +\infty} \Phi_{i \leftarrow j}(\tau; t). \tag{25}$$

From Eqs. (10) and (12), we obtain the differential equations for the probability density functions in the steady state as

$$\frac{d\tilde{I}_i(\tau)}{d\tau} = -\omega_{\text{rec}}(\tau)\tilde{I}_i(\tau), \tag{26}$$

$$\frac{d\tilde{S}_i(\tau)}{d\tau} = -\tilde{S}_i(\tau) \sum_{j=1}^{N} a_{ij}\tilde{\Phi}_{i \leftarrow j}(\tau). \tag{27}$$

In the steady state, the probabilities that a node is newly infected and that a node recovers are equal[11], so we have

$$\tilde{I}_i(0) = \tilde{S}_i(0). \tag{28}$$

In the steady state, the probabilities that node $i$ is in the infected state and in the susceptible state are, respectively,

$$\tilde{I}_i = \int_0^{+\infty} \tilde{I}_i(\tau)d\tau \tag{29}$$

and

$$\tilde{S}_i = \int_0^{+\infty} \tilde{S}_i(\tau)d\tau. \tag{30}$$

For type-I activation mechanism, we can obtain a relation between the probabilities of node $i$ being in the infected and susceptible state as

$$\frac{1}{\tilde{I}_i} = \frac{1}{\tilde{S}_i} \sum_{n=1}^{+\infty} \delta_{\text{eff}} \vartheta^{(n)}(0) \left( \frac{1}{\sum_{j=1}^{N} a_{ij}\tilde{I}_j} \right)^n, \tag{31}$$

or

$$\frac{1}{\tilde{I}_i} = \frac{1}{\lambda_{\text{eff}}^* \tilde{S}_i \sum_{j=1}^{N} a_{ij}\tilde{I}_j} + \frac{1}{\tilde{S}_i} \sum_{n=2}^{+\infty} \delta_{\text{eff}} \vartheta^{(n)}(0) \left( \frac{1}{\sum_{j=1}^{N} a_{ij}\tilde{I}_j} \right)^n, \tag{32}$$

where the effective recovery rate is

$$\delta_{\text{eff}} = \frac{1}{\int_0^{+\infty} \Psi_{\text{rec}}(\tau)d\tau}, \tag{33}$$

$\vartheta(\tau)$ is the inverse of the function $\Omega(\tau)$ given by

$$\Omega(\tau) = \delta_{\text{eff}} \int_0^{\tau} \int_0^{+\infty} \omega_{\text{inf}}[\min(\tau', \tau'')]\Psi_{\text{rec}}(\tau')d\tau'd\tau'', \tag{34}$$

and $\vartheta^{(n)}(0)$ is the $n$th derivative of $\vartheta(\tau)$ at $\tau = 0$ and $\lambda_{\text{eff}}^*$ is defined as

$$\lambda_{\text{eff}}^* = 1/[\delta_{\text{eff}}\vartheta^{(1)}(0)]. \tag{35}$$

The procedure to derive Eqs. (31) and (32) is detailed in Supplementary Note 1.

For type-II activation mechanism, the relation between the probabilities can be obtained as

$$\tilde{I}_i = \lambda_{\text{eff}} \tilde{S}_i \sum_{j=1}^{N} a_{ij}\tilde{I}_j, \tag{36}$$

where $\lambda_{\text{eff}}$ is the effective infection rate with the specific form

$$\lambda_{\text{eff}} = \int_0^{+\infty} \eta(\tau)\Psi_{\text{rec}}(\tau)d\tau. \tag{37}$$

Substituting Eq. (17) for $\eta(\tau)$ into Eq. (37), we have

$$\lambda_{\text{eff}} = \frac{1}{2\pi i} \int_C \frac{\hat{\psi}_{\text{inf}}(s)\hat{\psi}_{\text{rec}}(-s)}{1 - \hat{\psi}_{\text{inf}}(s)} \frac{ds}{s}, \tag{38}$$

where $\hat{\psi}_{\text{inf}}(s) = \int_0^{+\infty} \psi_{\text{inf}}(\tau)e^{-s\tau}d\tau$, $\hat{\psi}_{\text{rec}}(s) = \int_0^{+\infty} \psi_{\text{rec}}(\tau)e^{-s\tau}d\tau$, and $C$ is a contour that encloses the entire $\text{Re}(s) > 0$ region[42,64] (see Supplementary Note 2 for details).

Under what conditions can one expect the non-Markovian SIS dynamics to reduce to Markovian dynamics in the steady state? To address this question, we note that the equivalence requires that the non-Markovian steady-state equations be written[42,47] in the form of Eq. (36), or

$$\frac{1}{\tilde{I}_i} = \frac{1}{\lambda_{\text{eff}} \tilde{S}_i \sum_{j=1}^{N} a_{ij}\tilde{I}_j}. \tag{39}$$

For non-Markovian SIS spreading dynamics with type-II activation mechanism, such an equivalence does exist.

However, for processes with type-I activation mechanism, because of the high-order terms ($n \geq 2$) in Eq. (32), in general an equivalence to Markovian dynamics cannot be expected. Nonetheless, under certain conditions, an approximate equivalence can arise. In particular, if both $\vartheta^{(1)}(0)$ and $\vartheta^{(n)}(0)(n \geq 2)$ have a finite value, a large value of $\sum_{j=1}^{N} a_{ij}\tilde{I}_j$ will dominate the right-hand side of Eq. (32), making the high-order terms $\sum_{n=2}^{+\infty} \vartheta^{(n)}(0)(1/\sum_{j=1}^{N} a_{ij}\tilde{I}_j)^n$ negligibly small. The approximation depends on the local nodal properties, which is typically valid locally for large degree nodes and high infected density of neighboring nodes. For the small-degree

nodes, the high-order terms in Eq. (32) can no longer be neglected. In general, the validity of the approximation for the whole network depends not only on the infection density but also on the network structure, and the approximate equivalence holds if the small-degree nodes are not abundant in the network. If there is an appreciable fraction of small-degree nodes in a strongly hetero-geneous network, the approximate equivalence would fail. Other-wise, the effective infection rate can be approximated as

$$\lambda_{\text{eff}} \approx \lambda_{\text{eff}}^{*}. \tag{40}$$

Furthermore, when the infection process is Markovian regardless of whether the recovery process remains non-Markovian or Marko-vian, there exists an equivalence between non-Markovian and Markovian types of SIS dynamics with the following equality (see Supplementary Note 1):

$$\lambda_{\text{eff}} = \lambda_{\text{eff}}^{*}. \tag{41}$$

For non-Markovian SIS dynamics which can be equivalent to Markovian dynamics, the outbreak threshold is given by[65]

$$\lambda_{\text{c}}^{\text{eff}} = \frac{1}{\Lambda_{\text{max}}}, \tag{42}$$

where $\Lambda_{\text{max}}$ is the maximum eigenvalue of the network adjacency matrix. For processes with type-I mechanism where there is no equivalence between non-Markovian and Markovian dynamics, it is generally difficult to identify a relevant parameter to characterize the phase transition associated with disease outbreak.

To test when an equivalence between non-Markovian and Markovian spreading processes arises, we focus on the effects of the infection time distribution on the stationary infected density with both types of edge activation mechanisms. We first consider the case of type-II mechanism. We set $\alpha_{\text{I}} = 0.5, 1, 2, 4, \alpha_{\text{R}} = 2$, and $\beta_{\text{R}} = 0.5$ and adjust $\lambda_{\text{eff}}$ by changing $\beta_{\text{I}}$ through Eq. (37). Figure 4 shows the simulation results on random, scale-free and Hamsterster networks, which are the same as the networks in Fig. 3. It can be seen that, regardless of the network structure, the stationary infected density agrees well with both the result from Markovian process simulation and the analytical solution of the Markovian dynamics from Eq. (39) or (36). The theoretical thresholds calculated from Eq. (42) on random, scale-free, and Hamsterster networks are 0.096, 0.040 and 0.020, respectively, which are quite close to the simulated values of the threshold in Fig. 4. (Supplementary Fig. 5 in Supplementary Note 6 provides a comparison of stationary probability density distribution between simulation and theoretical prediction).

We now turn to the type-I case. For $\alpha_{\text{I}} = 1$, the distribution of infection time is exponential, making the underlying non-Markovian process completely equivalent to a Markovian process with the effective recovery rate $\delta_{\text{eff}}$ and effective infection rate $\lambda_{\text{eff}}$ given by Eqs. (33) and (41), respectively. Indeed, we find that, as $\lambda_{\text{eff}}$ is increased, or, equivalently, as $\beta_{\text{I}}$ is decreased, the infected density for the non-Markovian process with type-I activation mechanism agrees with the theoretical value of the corresponding Markovian process (Supplementary Fig. 6 in Supplementary Note 7 shows the results). For $\alpha_{\text{I}} \neq 1$, the exact equivalence breaks down. However, under certain conditions, an approximate equiva-lence can still be expected. To obtain the approximate equivalence, we analyze the effect of higher-order terms in Eq. (31) for non-Markovian process with type-I activation mechanism. For simplicity, we assume that $\psi_{\text{rec}}(\tau)$ is from

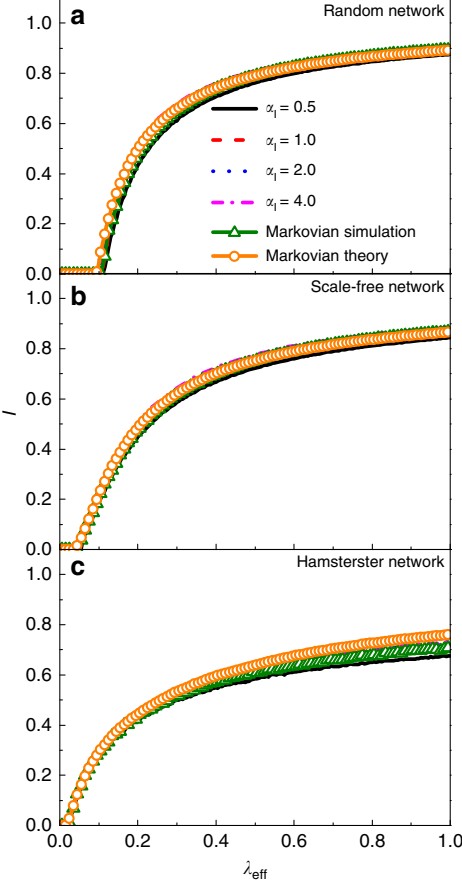

**Fig. 4** Demonstration of equivalence. Comparison of stationary infected density with type-II activation mechanism between non-Markovian and Markovian processes for **a** random **b** scale-free, and **c** Hamsterster networks which are the same as the networks in Fig. 3. Solid, dashed, dotted, and dot-dashed curves represent the results for $\alpha_{\text{I}} = 0.5, 1, 2, 4$, respectively, and the triangles and circles correspond to simulation results from the Markovian process and the analytical solutions, respectively. Other parameters are $\alpha_{\text{R}} = 2$ and $\beta_{\text{R}} = 0.5$

Eq. (22) but $\psi_{\text{inf}}(\tau)$ is a Beta distribution:

$$\frac{1}{B(\sigma, \gamma)} \kappa^{\sigma-1} (1 - \kappa)^{\gamma-1} \tag{43}$$

with $\sigma = 1$. We have

$$\psi_{\text{inf}}(\kappa) = \gamma (1 - \kappa)^{\gamma-1}, \tag{44}$$

where $0 \leq \kappa < 1$ and $\omega_{\text{inf}}(\kappa) = \gamma/(1 - \kappa)$. Especially, Eq. (44) turns into a uniform distribution $\psi_{\text{inf}}(\kappa) = 1 (0 \leq \kappa < 1)$ for $\gamma = 1$. We set $\alpha_{\text{R}} = 0.5, 1, 2, 4$ and $\beta_{\text{R}} = 0.5$, and adjust $\vartheta^{(n)}(0)$ by changing the value of $\gamma$. From Eqs. (34), (31) and (44), for a fixed value of $\alpha_{\text{R}}$, the proportional relationship among all $\vartheta^{(n)}(0)$ can remain invariant through a proper modification in the value of $\gamma$. In particular, an increase in the value of $\theta$ multiplying $\gamma$ will result in an increase in the value of $\omega_{\text{inf}}(\tau)$ by $\theta$ times. As a result, every value of $\Omega(\tau)$ will increase by the same factor $\theta$. For $\forall n \geq 0$, the quantity $\vartheta^{(n)}(0)$ will decrease by the factor of $\theta$, making the proportional relationship unchanged. From Eq. (40), we obtain the approximate effective infection rate as $\lambda_{\text{eff}} \approx \gamma/\delta_{\text{eff}}$. Figure 5 demonstrates a significant deviation in the stationary infected density, when its values are relatively small, of the

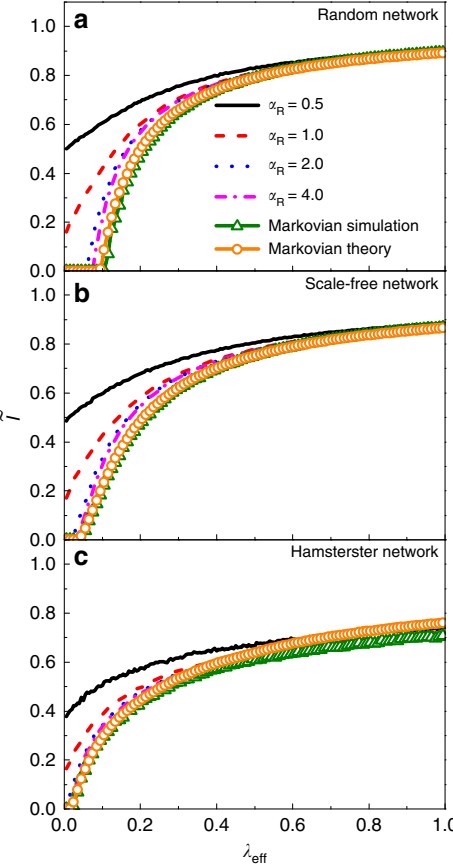

**Fig. 5** Demonstration of approximate equivalence. Shown are the stationary infected density with type-I activation mechanism for **a** random, **b** scale-free, and **c** Hamsterster networks, which are the same as the networks in Fig. 3, for $\beta_R = 0.5$. Solid, dashed, dotted, and dot-dashed curves represent the results for $\alpha_R = 0.5, 1, 2, 4$, respectively. Triangles and circles denote the results from the Markovian process and its analytical solution, respectively. An approximate equivalence arises in the regime of large infected density

non-Markovian process from that of the Markovian process with the effective infection rate $\lambda_{\text{eff}}$. However, as the values of $\lambda_{\text{eff}}$ and $\tilde{I}$ are increased, the results for the non-Markovian process gradually converge to those for the Markovian process. For a large value of $\alpha_R$, the values of $\vartheta^{(n)}(0)$ for $n \geq 2$ are reduced, decreasing the gap between the results from the non-Markovian and Markovian processes. Simulation results on strong heterogeneous networks also verify these results (see Supplementary Fig. 7 in Supplementary Note 8 for details).

## Discussion
Disease or virus spreading dynamics in networked systems in the real world are non-Markovian in that the temporal sequences of the occurrences of the key events underlying the spreading process do not follow a Poisson distribution. The non-Poisson behaviors make mathematical analyses of the spreading dynamics difficult, hindering their understanding. It is desired to develop a theoretical framework for non-Markovian spreading dynamics on complex networks. An issue of interest concerns about the inter-relation between non-Markovian and Markovian dynamics. Especially, while Markovian spreading processes with the Poisson characteristic are not a true reflection of real world situations, rigorous analyses and a relatively comprehensive understanding

of the underlying dynamics are possible, as demonstrated in the past two decades[5,7]. A curiosity driven and practically significant question is then under what conditions will a non-Markovian process be equivalent to a Markovian one. And, if such an equivalence does not exist, to what extent can a non-Markovian process be approximated by a Markovian one? In spite of previous work on non-Markovian spreading dynamics on complex networks[34,35,37,39–48], these issues have not been addressed satisfactorily, which motivated our current work.

We have developed a first-order mean field theory to solve both the transient phase and steady states of non-Markovian, SIS type of spreading dynamics. The theory can be used to assess accurately the difference between non-Markovian and Markovian dynamics, for any network structure. A finding is that, whether there is an equivalence between non-Markovian and Markovian processes depends on the specific edge activation mechanism. There are situations where non-Markovian SIS type of dynamics cannot be understood in terms of equivalent Markovian dynamics. We have identified one generic condition under which a complete equivalence between non-Markovian and Markovian processes holds: absence of any temporal correlation on active edges. When the correlation cannot be ignored due to the influence of susceptible nodes on active edges, the equivalence no longer holds. However, an approximate equivalence may still hold for the whole network, depending on the infection density and the network structure. We have found that, in the regime of relatively large infected density, a non-Markovian process can be approximated by a Markovian one for heterogeneous networks if small-degree nodes are not abundant in the network. All these findings were enabled by the theory developed in this paper.

While we have focused on non-Markovian, SIS type of spreading dynamics, a hope is that our mean-field theory can be extended to other types of dynamics such as SIR spreading or even cascading processes. Our work suggests the importance of accurately identifying the edge activation mechanisms responsible for spreading processes in the real world, which are key to determining whether an equivalence to Markovian dynamics exists so as to gain a deeper understanding of the underlying spreading process. Besides the two edge activation mechanisms, other types of edge activation mechanisms have been studied in the literature. For example, rule 2 in ref. [43] prescribes that the age of an active link is solely determined by the age of the infected node, which bears certain similarity but not identical to type-II edge activation studied in this paper. For this edge activation mechanism, an equivalence between non-Markovian and Markovian processes holds (see Supplementary Note 9 and Supplementary Figs. 8–10 for details). The issues of dynamical correlation and network communicability[66] are also critically important to non-Markovian spreading dynamics, for which a theoretical framework, e.g., a higher-order mean field framework, is lacking. An interesting question is whether the equivalence holds in the higher-order mean field framework or in a transient process. For example, when the system is in transient, the distribution of the state age varies with time, making invalid dimension reduction in the analysis of the infection and susceptible probabilities. At the present, it is not feasible to determine whether there is an equivalences between non-Markovian and Markovian processes in the transient regime.

## Methods
**Random number generation**. Taking $\psi_{\text{inf}}(\kappa)$ for example, its survival probability distribution is expressed as $\Psi_{\text{inf}}(\kappa) = \int_{\kappa}^{+\infty} \psi_{\text{inf}}(\kappa')d\kappa'$. In order to obtain a random number $\kappa$, we generate a random number $p$ uniformly distributed between zero and one and solve the equation

$$\Psi_{\text{inf}}(\kappa) = p. \tag{45}$$

For the Weibull distribution in Eq. (21) and the Beta distribution in Eq. (44), the value of $\kappa$ is given by

$$\kappa = \beta_1 (\ln \frac{1}{p})^{1/\alpha_1} \tag{46}$$

and

$$\kappa = 1 - p^{1/\gamma}, \tag{47}$$

respectively. We calculate the elapsed time $\tau$ with the distribution $\psi_{rec}(\tau)$ in a similar way.

**Simulation method**. In a network, each infected node or active edge is marked with an event-time defined as the absolute time when the infected node will recover or when the active edge will transmit the disease. The event-times of susceptible nodes and non-active edges are set as $+\infty$. All nodes and edges are assigned to a min-heap according to their event-times. Once the event-time of a node or an edge has changed, its position will be shifted in the min-heap instantaneously. Thus, the event-time of the node or edge at the root must be minimum. At each step, it is only necessary to make the node or edge at the root of the min-heap recover or transmit the disease[62], respectively.

For an infected node located at the root, meaning that the recovery event of the node will occur first among all the events including recovery and disease-transmitting events in the network, we update the absolute time $t$ to the event-time of the node and let the node recover. The event-time of the node then turns into $+\infty$. This recovery event will lead to state changes of some edges connected to the node. If some edges become active, we assign them new event-times $\kappa + t$, where $\kappa$ is a random number generated from the distribution $\psi_{inf}(\kappa)$ and $t$ is the current absolute time. The new event-time means the active edge will transmit disease at the absolute time $\kappa + t$. If some edges become non-active, their event-times become $+\infty$.

If an active edge is located at the root, i.e., the disease transmission event of the edge will occur first among all the events, we update the absolute time $t$ to the event-time of the edge, and then let the edge transmit disease. This transmission event can lead to a series of state changes of some nodes or edges and, consequently, the event-times of the new active and non-active edges are renewed to $\kappa + t$ and $+\infty$, respectively. In addition, the event-time of a new infected node will be updated to $\tau + t$, where $\tau$ follows the distribution $\psi_{rec}(\tau)$.

## Data availability
The source data underlying Figs. 3–5 and Supplementary Figs. 2–9 are provided as a Source Data file.

## Code availability
C++ code to reproduce the data in the main text and the Supplementary Information is available at https://github.com/fengmi9312/Codes-for-NCOMMS-19-00777.

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

## Acknowledgements

This work was supported by the National Natural Science Foundation of China under Grants No. 11575041, No. 11975099 and No. 61673086, the Natural Science Foundation of Shanghai under Grant No. 18ZR1412200, and the Science and Technology Commission of Shanghai Municipality under Grant No. 18dz2271000. YCL would like to acknowledge support from the Vannevar Bush Faculty Fellowship program sponsored by the Basic Research Office of the Assistant Secretary of Defense for Research and Engineering and funded by the Office of Naval Research through Grant No. N00014-16-1-2828.

## Author contributions

M.F. and M.T. designed research; M.F. performed research; S.-M.C., M.T., and Y.-C.L. contributed analytic tools; M.F., S.-M.C., M.T., and Y.-C.L. analyzed data; M.F., M.T., and Y.-C.L. wrote the paper.

## Additional information

**Competing interests:** The authors declare no competing interests.

