## [Peer Review File · Nature Communications]

Reviewers' comments:

Reviewer #1 (Remarks to the Author):

The paper entitled "Non-Markovian and Markovian spreading on networks: When are they equivalent?" introduces a first-order mean-field framework to analyse the evolution of a general non-Markovian susceptible-infected-susceptible model on networks. It tracks the probability density for each node to be in a certain state for a given period, leading to a set of partial differential equations. It is shown that the method reproduce with good accuracy the transient and stationary regime on different substrates. The framework is then used to show the approximate equivalence of non-Markovian and Markovian spreading in the stationary state. The approximate equivalence is accurate for certain infection rule, while for others it is also required to have a large infection density, and it seems to be dependent on the local properties of the nodes.

The mean-field framework introduced is interesting. It appears to be general enough to be applied to other types of spreading processes with multiple states. However, while the results concerning the approximate equivalence are also interesting, there are several issues that need to be addressed for them to be truly general and useful.

Major comments:

1. The Type-II edge activation mechanism is, up to my knowledge, new, and would require justifications for its introduction. The fact that the age of an active edge is reset upon the success (or failure) of a transmission is atypical of what is found in the literature. The authors cite Ref. [41] for the mechanisms. The Type-I edge activation do correspond to rule 1 in Ref. [41], however the Type-II edge activation does not correspond with rule 2. Rule 2 prescribes that the age of an active link is solely determined by (and corresponds to) the age of the infected node.

The mechanism used here could be considered to be yet another "memoryless" process, but with a non-Markovian rate between individual attempts. This is contrary to most non-Markovian models used in practice and therefore limits the applicability of the equivalence that follow from this mechanism. The authors should better justify the mechanism and discuss the relevant caveats.

2. The case of non-Markovian transmission based on rule 2 of Ref. [41] should be analysed. This is an interesting case that was studied, among others, by Refs. [39,41,45,46]. For instance, it is an appropriate mechanism to describe disease spreading with a significantly time-varying viral load for

infectious individuals (e.g. HIV). Moreover, the exact equivalence proved by Ref. [45] hinges upon this mechanism, a paper closely related to the proposed theoretical framework. An expected and desired properties for the proposed framework would be to recover, although approximately, this equivalence for the mean-field regime. This should not be too difficult given that this mechanism is simpler than the Type-II edge activation mechanism proposed in the current manuscript.

3. One of the key results is that the existence of the equivalence depends on the specific edge activation mechanism. It is claimed as a generic condition, but the framework makes use of a mean-field description. Do the authors have evidence or insights on why this should hold for the true process? Otherwise, the statements should be nuanced in the text.

4. Another concern is about the approximate equivalence for Type-I edge activation. It is claimed that one can use the effective infection rate if the sum of $a_{ij} I_j$ is large. However, this strongly depends on the local properties of node i . The worst-case scenario would be if node i has only one neighbour j , which is common for heterogeneous distributions, in which case terms of higher order n in Eq. (24) of the manuscript can no longer be ignored.

Therefore, it is important to mention that the approximate equivalence is only locally valid for certain nodes (typically with large degree). It appears that the validity of the approximation for the whole network will depend on the structure considered and not only on the infection density.

Minor comments:

1. Equations (19) to (26) are not self-explanatory : without a careful reading of the Supplementary Information (SI), it is difficult to interpret them. I would suggest putting most of them in the SI, or give a more detailed explanation of the terms involved.

2. Are the results for Fig. 4 and Fig. 5 based on the same networks used for Fig. 3? If so, it should be mentioned.

3. Is it time $t + dt$ that should be mentioned at the top of Fig. 2(b)?

4. One advantage of the framework proposed compared to other approaches, such as Ref. [40], is that it captures the transient regime. However, little is discussed about it, except to say that the formalism is able to capture the transient phase. Are there equivalences with Markovian processes even in this regime? That would make the transient results much more interesting.

Reviewer #3 (Remarks to the Author):

In this paper a mean-field theory is proposed for non-Markovian SIS epidemic dynamics on a network. Partial differential equations (similar to those found for fully-mixed populations subject to age-dependent infections in, for example, Hethcote, SIAM Review 42 (4), 599-653, 2000) are derived for the states of each node, and are solved numerically. In addition, analysis of the steady state behaviours enables a comparison of Markovian and non-Markovian cases, which have been found to be related in some previous literature.

The theoretical approach is novel and sheds light on an important question. I believe it will prove impactful within the community, where the focus on non-Markovian behaviour is growing. The paper is generally well-written and explains the assumptions of the theory reasonably well. The one lacking aspect is perhaps a strong new statement about non-Markovian aspects in SIS epidemics: I am thinking here of the speeding-up/slowing-down role of non-Markovianity in information-spread processes that is shown analytically in, for example, Iribarren and Moro, Phys. Rev. E, 84, 046116 (2011). The current paper invents a clever new method and applies it to check and help disambiguate existing results. This is a very valuable contribution, but for Nature Communications I would like to see the authors also point to any new results that haven't previously been shown by other approaches. Tracking transient behaviour itself is interesting, but as Fig 3(a) $\alpha=2$ (green) shows, the accuracy level is not always guaranteed to be high.

1. It isn't clear to me to what extent the "mean-field" approximation affects the accuracy of the results. If the authors use a purely Markovian case in their equations, presumably the equations reduce to known (approximate) mean-field SIS equations. A comment on situations where such mean-field equations are known to be inaccurate (examples are in Fig. 1 of Gleeson, Phys. Rev. Lett., 107, 068701 (2011), for example) would assist the reader in understanding possible limitations of the current theory; it would also be interesting to know if the non-Markovian approximations exhibit more or less inaccuracy than the corresponding Markovian ones. At present the implicit implication is that only the non-Markovian aspect of SIS on networks is problematic, while the truth is that even the Markovian case can cause problems for mean-field approaches.

2. Following Eq (7), the ages of two connected nodes are assumed to be uncorrelated. This approximation is exact for the Markovian case, but has any attempt been made to assess its impact in the non-Markovian case?
3. In Sec. II C a scale-free network is introduced as an important test case. However, the information given (number of nodes and mean degree) is too scanty to enable the reader to reproduce the results: the power-law exponent is not given, nor are the lower and upper cutoff degrees, which all play important roles in SIS dynamics. For a representative scale-free network, I would expect the power-law exponent to be less than 2.5, otherwise the role of hubs is underrepresented relative to real-world social networks.
4. End of Sec II C: it is claimed that random networks have longer transients due to the lack of hub nodes; can any evidence be offered for this claim? It is possible that any of the aspects mentioned in my previous point (in particular the lower cutoff degree) could also have an effect on the transient.
5. Following Eq (17): "In steady state the probabilities that a node is newly infects and that a node recovers are equal": this is presumably to balance fluxes so that the average number of new infections equals the average number of new recoveries, but the statement given seems stronger than that: is it indeed necessary that each node individually have the equal probabilities as in Eq (18) (rather than just the average over all nodes being equal)?
6. Surely Eq (33) is for Markovian networks (as in Ref 62), not for non-Markovian networks as claimed?
7. Sec IVA: "evenly distributed" would be better as "uniformly distributed"

Response to referee comments and description of changes made

Reviewer #1

General comments: *“The paper entitled ‘Non-Markovian and Markovian spreading on networks: When are they equivalent?’ introduces a first-order mean-field framework to analyse the evolution of a general non-Markovian susceptible-infected-susceptible model on networks. It tracks the probability density for each node to be in a certain state for a given period, leading to a set of partial differential equations. It is shown that the method reproduce with good accuracy the transient and stationary regime on different substrates. The framework is then used to show the approximate equivalence of non-Markovian and Markovian spreading in the stationary state. The approximate equivalence is accurate for certain infection rule, while for others it is also required to have a large infected density, and it seems to be dependent on the local properties of the nodes.*

The mean-field framework introduced is interesting. It appears to be general enough to be applied to other types of spreading processes with multiple states. However, while the results concerning the approximate equivalence are also interesting, there are several issues that need to be addressed for them to be truly general and useful.”

Response: We appreciate that the referee considered our mean-field framework “interesting” and “general enough,” and raised a number of issues to make our work “truly general and useful.” We have fully addressed these issues, as described below.

Major Comment 1: *“The Type-II edge activation mechanism is, up to my knowledge, new, and would require justifications for its introduction. The fact that the age of an active edge is reset upon the success (or failure) of a transmission is atypical of what is found in the literature. The authors cite Ref. [41] for the mechanisms. The Type-I edge activation do correspond to rule 1 in Ref. [41], however the Type-II edge activation does not correspond with rule 2. Rule 2 prescribes that the age of an active link is solely determined by (and corresponds to) the age of the infected node.*

The mechanism used here could be considered to be yet another “memoryless” process, but with a non-Markovian rate between individual attempts. This is contrary to most non-Markovian models used in practice and therefore limits the applicability of the equivalence that follow from this mechanism. The authors should better justify the mechanism and discuss the relevant caveats.

Response: The referee is completely correct that type-II edge activation in our manuscript is not the same as rule #2 in Ref. [43] (i.e., Ref. [41] in previous version). In fact, this type of edge activation corresponds to the mechanism in Ref. [47] (i.e., Ref. [45] in previous version) (see Fig. 1 therein). We have consulted the main author of Refs. [43] (i.e., Ref. [41] in previous version) and [47] (i.e., Ref. [45] in previous version), and confirmed that the type-II edge activation described in our manuscript does correspond to the mechanism in Ref. [47] (i.e., Ref. [45] in previous version). Since we focus on the effects of non-Markovian *interevent* times on spreading dynamics, it is reasonable to assume that the value of the time elapsed T_{AB} will turn to zero again

when node A transmits a disease to node B , which is a modified version of rule #2 in Ref. [43] (i.e., Ref. [41] in previous version).

We have revised the corresponding description in the last paragraph of left column on page 3. The original phrases “For the first mechanism (type-I)” and “For the second mechanism (type-II)” have been changed to “For the first mechanism (type-I), which is the same as rule #1 in Ref. [43]” and “For the second mechanism (type-II), which is the same as that in Ref. [47],” respectively.

Major comment 2: *“The case of non-Markovian transmission based on rule 2 of Ref. [41] should be analysed. This is an interesting case that was studied, among others, by Refs. [39, 41, 45, 46]. For instance, it is an appropriate mechanism to describe disease spreading with a significantly time-varying viral load for infectious individuals (e.g. HIV). Moreover, the exact equivalence proved by Ref. [45] hinges upon this mechanism, a paper closely related to the proposed theoretical framework. An expected and desired properties for the proposed framework would be to recover, although approximately, this equivalence for the mean-field regime. This should not be too difficult given that this mechanism is simpler than the Type-II edge activation mechanism proposed in the current manuscript.”*

Response: As stated in our Response to Comment 1, type-II edge activation in our work corresponds to the mechanism in Ref. [47] (i.e., Ref. [45] in previous version). In the stationary state, there is an equivalence between non-Markovian SIS and the corresponding Markovian dynamics. Rule #2 in Ref. [43] (i.e., Ref. [41] in previous version) prescribes that the age of an active link is solely determined by (and corresponds to) the age of the infected node, which is not identical to type-II edge activation described in our manuscript.

Following referee’s suggestion, we have studied the dynamics associated with rule #2 in Ref. [43] (i.e., Ref. [41] in previous version), which we now call type-III edge activation mechanism in the revised manuscript. The analyses and simulations have been included in Sec. IX of Supplementary Information. In the main text (Sec. III - Discussion), we have added the following explanation:

- Besides the two edge activation mechanisms, other types of edge activation mechanisms have been studied in the literature. For example, rule #2 in Ref. [43] prescribes that the age of an active link is solely determined by the age of the infected node, which bears certain similarity but not identical to type-II edge activation studied in this paper. For this edge activation mechanism, an equivalence between non-Markovian and Markovian processes holds (see details in Sec. IX of Supplementary Information).

Major comment 3: *“One of the key results is that the existence of the equivalence depends on the specific edge activation mechanism. It is claimed as a generic condition, but the framework makes use of a mean-field description. Do the authors have evidence or insights on why this should hold for the true process? Otherwise, the statements should be nuanced in the text.”*

Response 3: The equivalence holds in the first-order mean field framework. Whether such an equivalence still exists in real world processes requires further study. The statement has been nuanced in the current version. In particular, the original statement in the last paragraph of Sec. III (Discussion): “The issue of dynamical correlations and network communicability [63] is also critically important to non-Markovian spreading process, for which a theoretical framework

is still missing,” has been replaced by

- The issues of dynamical correlation and network communicability [66] are also critically important to non-Markovian spreading dynamics, for which a theoretical framework, e.g., a higher-order mean field framework, is lacking. An interesting question is whether the equivalence holds in the higher-order mean field framework or in a transient process.

Major comment 4: *“Another concern is about the approximate equivalence for Type-I edge activation. It is claimed that one can use the effective infection rate if the sum of $\alpha_{ij} I_j$ is large. However, this strongly depends on the local properties of node i . The worst-case scenario would be if node i has only one neighbour j , which is common for heterogeneous distributions, in which case terms of higher order n in Eq. (24) of the manuscript can no longer be ignored.*

Therefore, it is important to mention that the approximate equivalence is only locally valid for certain nodes (typically with large degree). It appears that the validity of the approximation for the whole network will depend on the structure considered and not only on the infected density.”

Response 4: The referee is absolutely right that the approximate equivalence in Eq. (20) (i.e., Eq. (24) in previous version) depends strongly on the local properties of node i . In a strongly heterogeneous network, the existence of many small-degree nodes can make the approximate equivalence fail. Following referee’s suggestion, in the left column on page 7, we have added the following explanation:

- The approximation depends on the local nodal properties, which is typically valid locally for large degree nodes and high infected density of neighboring nodes. For the small degree nodes, the high-order terms in Eq. (20) can no longer be neglected. In general, the validity of the approximation for the whole network depends not only on the infection density but also on the network structure, and the approximate equivalence holds if the small degree nodes are not abundant in the network. If there is an appreciable fraction of small-degree nodes in a strongly heterogeneous network, the approximate equivalence would fail.

In Sec. III (Discussion), the original statement “However, in this case, we have found that, in the regime of relatively large infection density, a non-Markovian process can be approximated by a Markovian one” has been replaced by

- However, an approximate equivalence may still hold for the whole network, depending on the infection density and the network structure. We have found that, in the regime of relatively large infected density, a non-Markovian process can be approximated by a Markovian one for heterogeneous networks if small degree nodes are not abundant in the network.

Minor comment 1: *“Equations (19) to (26) are not self-explanatory: without a careful reading of the Supplementary Information (SI), it is difficult to interpret them. I would suggest putting most of them in the SI, or give a more detailed explanation of the terms involved.*

Response: Following referee’s suggestion, we have moved Eqs. (19), (25) and (26) in the previous version into Supplementary Information as Eqs. (I. 11), (I. 19) and (I. 20). The original Eqs. (20)-(24) constitute part of the conclusion about the steady state under type-I edge activation mechanism, which are now Eqs. (19)-(23) in the revised manuscript.

Minor comment 2: *“Are the results for Fig. 4 and Fig. 5 based on the same networks used for Fig. 3? If so, it should be mentioned.”*

Response 2: Yes, the results for Figs. 4 and 5 are based on the same networks used for Fig. 3. This has now been explicitly stated in the captions of Figs. 4 and 5.

Minor comment 3: *“Is it time $t + dt$ that should be mentioned at the top of Fig. 2(b)?”*

Response: Figs. 2(b) and 2(c) show the infection and recovery processes in the transition period ($t, t+dt$). We have now specified “time of transition” in Figs. 2(b), 2(c), and Supplementary Figs. 1(b), 1(c).

Minor comment 4: *“One advantage of the framework proposed compared to other approaches, such as Ref. [40], is that it captures the transient regime. However, little is discussed about it, except to say that the formalism is able to capture the transient phase. Are there equivalences with Markovian processes even in this regime? That would make the transient results much more interesting.”*

Response: Whether there is an equivalences between non-Markovian and Markovian processes in the transient regime is a significant issue. When the system has reached a steady state, a node’s state age follows a stable distribution, so the probability that a node is in the infected or susceptible state can be readily obtained. However, when the system is in a transient, the distribution of the state age varies with time, making invalid dimension reduction in the analysis of the infection and susceptible probabilities. It is thus not generally feasible to determine whether there is an equivalence between non-Markovian and Markovian processes in the transient regime. We have provided some explanatory remarks at the end of the Discussion section.

In Supplementary Information, we have presented a more detailed discussion about the dynamics in the transient regime, including a second-order mean field approach and an error analysis (Sec. III). Specifically, the second order mean field approach includes and treats the dynamical correlation between a node and its neighbors, making the predictions more consistent with the simulation results but leading to a higher computational complexity. In fact, the dynamical correlations are a key factor affecting the accuracy of the first-order mean field prediction.

Reviewer #3

General comment: *“In this paper a mean-field theory is proposed for non-Markovian SIS epidemic dynamics on a network. Partial differential equations (similar to those found for fully-mixed populations subject to age-dependent infections in, for example, Hethcote, SIAM Review 42 (4), 599-653, 2000) are derived for the states of each node, and are solved numerically. In addition, analysis of the steady state behaviours enables a comparison of Markovian and non-Markovian cases, which have been found to be related in some previous literature.*

The theoretical approach is novel and sheds light on an important question. I believe it will prove impactful within the community, where the focus on non-Markovian behaviour is growing. The paper is generally well-written and explains the assumptions of the theory reasonably well. The one lacking aspect is perhaps a strong new statement about non-Markovian aspects in SIS epidemics: I am thinking here of the speeding-up/slowing-down role of non-Markovianity in information-spread processes that is shown analytically in, for example, Iribarren and Moro, Phys. Rev. E, 84, 046116 (2011). The current paper invents a clever new method and applies it to check and help disambiguate existing results. This is a very valuable contribution, but for Nature Communications I would like to see the authors also point to any new results that haven't previously been shown by other approaches. Tracking transient behaviour itself is interesting, but as Fig 3(a) $\alpha=2$ (green) shows, the accuracy level is not always guaranteed to be high.”

Response: We thank the referee for regarding that our work is “novel and sheds light on an important question”, it “will prove impactful within the community, where the focus on non-Markovian behaviour is growing”, and the paper “is generally well-written and explains the assumptions of the theory reasonably well”. To address referee’s general comment, we have developed a second-order mean field approach that generally gives better theoretical predictions as compared with the first-order theory in the original manuscript. We have also obtained new results (to be detailed below). The references pointed out by the referee [Phys. Rev. Lett. 103, 038702 (2009) and Phys. Rev. E 84, 046116 (2011)] are extremely relevant and valuable to our analysis, which have been cited as Refs. [36] and [38] in the main text (Refs. [17] and [18] in Supplementary Information).

Comment 1: *“It isn't clear to me to what extent the ‘mean-field’ approximation affects the accuracy of the results. If the authors use a purely Markovian case in their equations, presumably the equations reduce to known (approximate) mean-field SIS equations. A comment on situations where such mean-field equations are known to be inaccurate (examples are in Fig. 1 of Gleeson, Phys. Rev. Lett., 107, 068701 (2011), for example) would assist the reader in understanding possible limitations of the current theory; it would also be interesting to know if the non-Markovian approximations exhibit more or less inaccuracy than the corresponding Markovian ones. At present the implicit implication is that only the non-Markovian aspect of SIS on networks is problematic, while the truth is that even the Markovian case can cause problems for mean-field approaches.”*

Response: A more accurate theoretical approach describing the spreading dynamics (beyond the first-order mean field theory) requires that both the network topology and dynamical correlation be taken into account, at the expense of increasing complexity of simulation and analysis.

Following referee’s suggestion, we have proposed and studied a more comprehensive framework: a second-order mean field theory to address the effects of dynamical correlation on the network (Sec. III of Supplementary Information). In the new theory, the ages of two connected nodes are correlated, in contrast to the assumption used in the first order theory. Within the feasibility of computation and analysis, the new theoretical predictions have been verified for homogeneous networks. For example, Supplementary Fig. 2 shows that the second-order mean field theory can predict the simulation results more accurately than the first-order theory, even in some extreme situations [e.g., $\alpha_I = 4$ as shown in Supplementary Figs. 2(c) and 2(d)].

The highly relevant reference pointed out by the referee [Phys. Rev. Lett. 107, 068701 (2011)] has been cited with the following explanation (in the third paragraph on page 1):

- An approximate master equation theory can yield more accurate theoretical results and can be reduced to the pair-wise approximation theory and the mean field theory through some proper approximations [55].

In Sec. IIC, we have added the following explanation:

- However, in some cases (e.g., $\alpha_I = 4$ for ER random networks), there are some discrepancies between the predictions from the first-order mean-field theory and the simulation results [c.f., insets in Figs. 3(a) and 3(d)], due to the exclusion of any dynamical correlation. For a more accurate description of non-Markovian spreading processes, the dynamical correlation in the evolution of states of connected nodal pairs must be taken into account. To meet this challenge, we articulate a second-order mean field approach, and show that it is capable of predicting the simulation results more accurately than the first-order theory, even for extreme situations where the disease decays rapidly [e.g., the $\alpha_I = 4$ case in Supplementary Figs. 2(c) and 2(d)]. The second-order theory indicates that, in general, dynamical correlation such as temporal correlation between active edges can significantly affect the accuracy of the mean-field analysis (see Sec. III of Supplementary Information for details). Due to the limitation of computational feasibility, we have tested the predictions of the second-order mean field theory for homogeneous networks. To extend the study to heterogeneous networks is currently infeasible due to the extreme high computational complexity required to treat all possible nodal pairs separately.

In the last paragraph of Sec. III (Discussion), we have further addressed the issue of dynamical correlation:

- The issues of dynamical correlation and network communicability [66] are also critically important to non-Markovian spreading processes, for which a theoretical framework, e.g., a higher-order mean field framework, is lacking. An interesting question is whether the equivalence holds in the higher-order mean field framework or in a transient process.

Comment 2: *“Following Eq (7), the ages of two connected nodes are assumed to be uncorrelated. This approximation is exact for the Markovian case, but has any attempt been made to assess its impact in the non-Markovian case?”*

Response: For non-Markovian processes, the approximation to ignore the correlation will compromise the accuracy of the first-order theory. To overcome this difficulty, in the revised manuscript we have proposed a second-order mean field theory to take into account the age correlation between any pair of connected nodes. As shown in Supplementary Fig. 2, the second-order theory gives more accurate prediction, where the impact of age correlation on the accuracy can be assessed through an error analysis.

Comment 3: *“In Sec. II C a scale-free network is introduced as an important test case. However, the information given (number of nodes and mean degree) is too scanty to enable the reader to reproduce the results: the power-law exponent is not given, nor are the lower and upper cutoff degrees, which all play important roles in SIS dynamics. For a representative scale-free network, I would expect the power-law exponent to be less than 2.5, otherwise the role of hubs is underrepresented relative to real-world social networks.”*

Response: In the main text, we used the Barabási-Albert (BA) type of scale-free networks, where the power-law exponent is three, the minimum degree is five, and the upper cutoff degree is about 260. The information has been included in the caption of Fig. 3.

In Sec. V of Supplementary Information, we investigate the effects of the power-law degree exponent and the lower cutoff degree on the transient dynamics. Considering that, for the SIS model, the time that the system has reached a steady state cannot be defined precisely, we set a numerical rule to determine the transient time T_{half} - the time at which the infected density reaches the average between the initial and final steady-state density. We then fix the lower cutoff degree to be five to find the effect of the power-law exponent, and set the power-law exponent to be 2.3 to study the impact of varying the lower cutoff degree. In the last paragraph of Sec. IIC entitled “Transient behaviors,” we have added the following explanation:

- We further investigate the effects of infection time distribution and degree distribution on the transient time in Secs. IV and V of Supplementary Information, respectively.
- We find that a smaller value of the power-law exponent and a larger value of the lower cutoff degree can lead to a shorter transient lifetime, indicating that hub nodes in a scale-free network can expedite the spreading processes of the non-Markovian type. Supplementary Fig. 4 presents systematic results on the transient lifetime versus the values of the power-law exponent and the lower-cutoff degree.

We have also tested our theory on scale-free networks with a stronger heterogeneous degree distribution, where the power-law exponent is 2.3. As shown in Supplementary Fig. 7 of Sec. VIII of Supplementary Information, the theory is able to predict the simulation results well. For such networks, the approximate equivalence with type-I activation mechanism and the equivalence with type-II mechanism in steady state hold. The following statement has been added in the main text (Sec. IID2):

- Simulation results on strong heterogeneous networks also verify these results (see Sec. VIII of Supplementary Information for details).

Comment 4: “*End of Sec II C: it is claimed that random networks have longer transients due to the lack of hub nodes; can any evidence be offered for this claim? It is possible that any of the aspects mentioned in my previous point (in particular the lower cutoff degree) could also have an effect on the transient.*”

Response: We have studied the effects of the values of the power-law exponent and the lower cutoff degree on the transient dynamics, and found that a smaller value of the power-law exponent and a higher value of the low cutoff degree will speed up the spreading process. This indicates that hub nodes tend to lead to shorter transient lifetime (see our Response to Comment 3 and Sec. V in Supplementary Information).

Comment 5: “*Following Eq (17): ‘In steady state the probabilities that a node is newly infects and that a node recovers are equal’: this is presumably to balance fluxes so that the average number of new infections equals the average number of new recoveries, but the statement given seems stronger than that: is it indeed necessary that each node individually have the equal probabilities as in Eq (18) (rather than just the average over all nodes being equal)?*”

Response: In a simplified mean field theory, all nodes are assumed to be equivalent and they have the same probability of being infected. We can then have the initial condition for all nodes: $\tilde{I}(0) = \tilde{S}(0)$. In the heterogeneous mean field theory, all the nodes with the same degree are considered to be equivalent, henceforth the conditions $\tilde{I}_k(0) = \tilde{S}_k(0)$, where the subscript k denotes the degree. The assumption was described in the Review Article [Rev. Mod. Phys. 87, 925 (2015), e.g., in Eqs. (26) and (82)]. In the theoretical development of our work, we treat the individual node separately as in a quench mean field approach, and each node i has its own steady condition: $\tilde{I}_i(0) = \tilde{S}_i(0)$. This ansatz was used in previous studies [e.g., Computing 93, 147 (2011)], which has been cited as Ref. [11] after Eq. (17).

Comment 6: “*Surely Eq (33) is for Markovian networks (as in Ref 62), not for non-Markovian networks as claimed?*”

Response: If a non-Markovian spreading process is equivalent to a Markovian one, Eq. (24) (i.e., Eq. (27) in previous version) or Eq. (27) (i.e., Eq. (30) in previous version) will be satisfied, which is the steady state equation of Markovian spreading. A non-Markovian and a Markovian spreading process with the same value of λ_{eff} will then have the steady state with the same infected density, which means they will possess the identical threshold value. Eq. (30) (i.e., Eq. (33) in previous version) is also for this kind of non-Markovian spreading dynamics.

Comment 7: “*Sec IV. A: ‘evenly distributed’ would be better as ‘uniformly distributed’.*”

Response: This has been implemented in the revised manuscript.

REVIEWERS' COMMENTS:

Reviewer #1 (Remarks to the Author):

The authors provided a satisfying answer to my previous comments and have, I think, greatly improved their manuscript in doing so.

That being said, I also believe Reviewer 3 raised an important point regarding the lack of new important results beyond the methodology itself. I mention this because I wished I had also raised that issue in my initial review, and I am unsure that including a second-order mean field theory is sufficient. I am afraid that the manuscript might fail to convince new readers of why this new methodology is important.

However, based on the strength of the analytical work, I am happy to recommend publication, provided that Reviewer 3 is satisfied with the authors' answer about the novelty of the results.

Reviewer #3 (Remarks to the Author):

The authors have addressed all my points in a satisfactory manner and I am happy to recommend publication.

Response to Referees

July 26, 2019

Reviewer #1

General comments: *“The authors provided a satisfying answer to my previous comments and have, I think, greatly improved their manuscript in doing so.*

That being said, I also believe Reviewer 3 raised an important point regarding the lack of new important results beyond the methodology itself. I mention this because I wished I had also raised that issue in my initial review, and I am unsure that including a second-order mean field theory is sufficient. I am afraid that the manuscript might fail to convince new readers of why this new methodology is important.

However, based on the strength of the analytical work, I am happy to recommend publication, provided that Reviewer 3 is satisfied with the authors’ answer about the novelty of the results.”

Response: We appreciate the referee’s valuable comments and thank the referee for recommending publication.

Reviewer #3

General comment: *“The authors have addressed all my points in a satisfactory manner and I am happy to recommend publication.”*

Response: We thank the referee for recommending our paper to publication.